# Identifying Lightning Processes in ERA5 Soundings with Deep Learning

Gregor Ehrensperger[1, 2], Thorsten Simon[2], Georg J. Mayr[2], and Tobias Hell[1]

[1]Data Lab Hell GmbH, Austria
[2]Department of Atmospheric and Cryospheric Sciences, University of Innsbruck

**Correspondence:** Gregor Ehrensperger (gregor.ehrensperger@uibk.ac.at)

**Abstract.** Atmospheric environments favorable for lightning and convection are commonly represented by proxies or parameterizations based on expert knowledge such as CAPE, wind shear, charge separation, or combinations thereof. Recent developments in the field of high resolution reanalyses, accurate lightning observations, machine learning (ML) and explainable artificial intelligence (XAI) open possibilities for identifying tailored proxies without prior expert knowledge.

5     This study utilizes a deep neural network trained to match temporally and vertically well-resolved ERA5 soundings of cloud physics, mass-field, and wind-field variables with lightning observations from the *Austrian Lightning Information & Detection System* (ALDIS). The ML model only receives the raw model atmosphere data as inputs, without incorporating any expert parameters or proxies derived from the model levels. Using and adapting appropriate XAI methods, it is then demonstrated how the inner workings of this well-performing deep learning model can be uncovered to identify physically meaningful 10  patterns within the ERA5 soundings that describe lightning processes.

    The ERA5 parameters are taken on model levels beyond the tropopause forming an input layer of approx. 670 features, the lightning data are transformed to a binary target variable labeling the spatio-temporal ERA5 grid cells as *cells with lightning activity* and *cells without lightning activity*.

    Scaled SHAP values are introduced to highlight the atmospheric processes learned by the neural network and show that 15  the model identifies cloud ice and snow content in the upper and mid-troposphere as very relevant features. As these patterns correspond to the separation of charge in thunderstorm clouds, the deep learning model can serve as a physically meaningful description of lightning. The scaled SHAP values also reveal that, depending on the location, the model additionally learns to correctly classify cells with lightning activity by exploiting mass-field or wind-field variables.

    This approach also showcases how XAI can be used to accelerate knowledge discovery in areas where expert knowledge is 20  still scarce.

## 1 Introduction

Lightning affects many fields of our everyday's life. Cloud-to-ground flashes might hit infrastructure such as wind turbines (Becerra et al., 2018) and power lines (Cummins et al., 1998) and thus cause power outages. Humans might get injured (Ritenour et al., 2008) or even die (Holle, 2016) after being hit by lightning. Wildfires (Reineking et al., 2010) release carbon

dioxide into the climate system, and thus limit the biosphere's capacity to store carbon dioxide. Lightning also affects the climate system by producing nitrogen oxides which play a key role in ozone conversion and acid rain production (DeCaria et al., 2005). Ozone is an important greenhouse gas and changes in concentration can lead to warming or cooling of the atmosphere. Thus, understanding of lightning is also an important factor in climate change research (Finney et al., 2018).

Given lightning's impact and that an average of 46 flashes are occurring around the globe every second (Cecil et al., 2014) it is desirable to have models of the atmosphere capable to simulate lightning and its underlying dynamic processes down to the resolved scales of the numeric model. Beyond the resolved scales one relies on so called proxies *or* parameterizations to further describe lightning. The term *proxy* is commonly used for quantities derived from atmospheric model output *after* the simulation has run. *Parameterizations* diagnose lightning *while* the model is running and hence can feed back on the simulation.

Proxies are frequently applied to assess historic and future behavior of convection and lightning. Popular proxies are cloud top height (Price and Rind, 1992), cloud ice flux (Finney et al., 2014), CAPE times precipitation (Romps et al., 2018), or the lightning potential index (Brisson et al., 2021). Though, these proxies perform reasonably well (Tippett et al., 2019), there is a need for more complex or holistic proxies, as the behavior of lightning in a changing climate is still uncertain (Murray, 2018). Another application highlighting the need for further research on lightning description is operational weather forecasting. Experience indicates, for instance, that CAPE needs to be adapted to local conditions in order to perform well (Groenemeijer et al., 2019).

Parameterizations are an internal part of numeric models, as they emulate sub-scale processes that cannot be resolved due the discretization of governing equations. Therefore, the emulated processes give feedback to the other processes, also on larger scales, within the atmospheric model. For instance, Tost et al. (2007) showed that modeled nitrogen oxide is sensitive to lightning parameterizations in numerical models. Next to the classic description of lightning using cloud top height (Price and Rind, 1992), parameterizations have been developed using polynomial regression (Allen and Pickering, 2002) and schemes based on hydrometeors in the mixed-phase region which is important for cloud-resolving models (McCaul et al., 2009). A comparison of several parameterizations using a superparameterized model is given by Charn and Parishani (2021). Recently, the ECMWF launched a product for total lightning densities expressed as a function of hydrometeors contents, CAPE, and (convective) cloud-base height output by the convective parameterization (Lopez, 2016).

In recent years, also machine learning approaches have been proposed to describe convection and lightning. Forty preselected single-level parameters from ERA5 were processed by artificial neural networks and gradient boosting machines to study lightning in parts of Europe and Sri Lanka (Ukkonen et al., 2017; Ukkonen and Mäkelä, 2019). Other studies evaluated random forests for regions such as the Hubei Province in China (Shi et al., 2022) or the Southern Great Plains (Shan et al., 2023) and generalized additive models (GAM) for the European Alps (Simon et al., 2023). All these studies confirm that the use of ML approaches for the description of lightning is promising.

Very recently, *explainable artificial intelligence* (XAI) techniques are used to move towards understanding the underlying reasoning of complex AI models and show encouraging results in various Earth System Sciences applications (Barnes et al., 2020; Dutta and Pal, 2022; Hilburn et al., 2021; Mayer and Barnes, 2021; Stirnberg et al., 2021; Toms et al., 2021). Specifically, Silva et al. (2022) use XGBoost classification trees to explore when the NASA Goddard Earth Observing System model of

lightning flash occurrence shows weaknesses and apply *Shapley additive explanations* (SHAP) to describe which meteorological drivers are related to the model errors. They found that these errors are strongly related to convection in the atmosphere and certain characteristics of the land surface.

This paper builds upon these studies and demonstrates the use of explainable artificial intelligence to discover potential proxies favorable for lighting directly from raw model level atmospheric data. Unlike prior research (Ukkonen et al., 2017; Ukkonen and Mäkelä, 2019; Shi et al., 2022; Shan et al., 2023; Simon et al., 2023) that applied machine learning to classify lightning occurrence using preselected proxies derived from atmospheric parameters by experts, this work directly exploits the raw ERA5 model level data and targets at finding such proxies. Using model level data directly offers two key benefits. First, it reduces the risk of overlooking potentially significant atmospheric conditions that could be missed when concentrating solely on preselected proxies. Second, it provides a comprehensive view of the vertical atmospheric layers, requiring less meteorological expertise to prepare the input data. This approach, however, increases the dimensionality of the input layer with highly correlated features along the vertical axis, making commonly used feature importance graphs hard to interpret. Inspired by the use of SHAP values in imaging tasks, this work employs SHAP values to reason on model levels directly. Due to the high dimensionality of the input, out of the box plotting routines are not feasible for interpreting SHAP values in this context. Therefore, the obtained SHAP values are aggregated to provide a more global understanding of a feature's contribution to the final model output. To improve explainability, *scaled SHAP values* are introduced to align the SHAP values across all grid cells. The median, as well as the 25th and 75th percentiles of these scaled SHAP values are then visualized along the vertical profiles, aiding the interpretation of the patterns exploited by the model.

This study focuses on lightning during the peak phase of the warm season (June, July, August) which differs fundamentally in the underlying dynamic processes to lightning during the cold season (Morgenstern et al., 2022).

The region of interest are the Eastern Alps which are characterized by complex terrain. Atmospheric dynamics on a gamut of scales interact with topography, leading to various meso-scale processes (Feldmann et al., 2021) and local processes (Houze, 2012) that can trigger convection and lightning.

This paper is structured as follows. Section 2 presents both the lightning detection data and the atmospheric reanalyses. Section 3 describes the two modelling approaches and elaborates on the XAI method used to interpret the patterns identified by the deep learning model. The results of these analyses are given in Section 4. Section 5 discusses the physical patterns identified by the methods, highlights future applications and finally concludes the study.

## 2 Data

Two data sets build the foundation for this supervised machine learning task. First, the observational data from the lightning location system ALDIS (Section 2.1) is used to derive the labels distinguishing cells with and without lightning activity. Second, pseudo soundings from ERA5 (Section 2.2) serve as input for the deep learning approach. Spatially, the grid centers range from $8.25°E$ to $16.75°E$ and from $45.25°N$ to $49.75°N$.

**Table 1.** ERA5 parameters on model levels.

| Name | Short Name | Units | Parameter ID |
|---|---|---|---|
| Temperature | t | K | 130 |
| Specific humidity | q | $\mathrm{kg\,kg^{-1}}$ | 133 |
| U component of wind | u | $\mathrm{m\,}s^{-1}$ | 131 |
| V component of wind | v | $\mathrm{m\,}s^{-1}$ | 132 |
| Vertical velocity | w | $\mathrm{Pa\,s^{-1}}$ | 135 |
| Specific rain water content | crwc | $\mathrm{kg\,kg^{-1}}$ | 75 |
| Specific snow water content | cswc | $\mathrm{kg\,kg^{-1}}$ | 76 |
| Specific cloud liquid water content | clwc | $\mathrm{kg\,kg^{-1}}$ | 246 |
| Specific cloud ice water content | ciwc | $\mathrm{kg\,kg^{-1}}$ | 247 |

Temporally, data for the meteorological summers (June, July, August) from 2010 to 2019 are available. The data of 2010–2018 serve as training/validation[1], the data from 2019 is reserved as truly independent test data.

## 2.1 Lightning Detection Data

The Austrian Lightning Detection & Information System (ALDIS) is part of the European Cooperation for Lightning Detection (EUCLID) (Schulz et al., 2016). Cloud-to-ground flashes with a current greater than $15\,\mathrm{kA}$ or smaller than $-2\,\mathrm{kA}$ are aggregated to the spatio-temporal grid cells of ERA5 (Section 2.2). Each cell has a horizontal extent of $0.25° \times 0.25°$ and temporally of one hour. If at least one flash has been detected in such a grid cell, then the cell is labeled as *cell with lightning activity*. Otherwise, if not a single flash has been detected, the cell is labeled as *cell without lightning activity*.

## 2.2 Atmospheric Reanalysis

ECMWF's fifth reanalyses, ERA5 (Hersbach et al., 2020), is available at a horizontal resolution of $0.25° \times 0.25°$ (in the region of interest this corresponds to approx. $19\,\mathrm{km} \times 28\,\mathrm{km}$) and temporally of one hour. Vertically it consists of 137 hybrid model levels that align with topography near ground and approach isobars in the upper atmosphere[2]. On these model levels nine parameters (Table 1) are available to describe the state of the atmosphere. In addition to classical parameters such as temperature, specific humidity and three-dimensional winds, ERA5 provides a description of liquid and solid water particles in clouds, i.e. the specific content of ice, snow (including graupel), liquid water, and rain. For this study, these parameters are used on the lowest 74 model levels, spanning from level 64 (approx. $15\,000\,\mathrm{m}$ geopotential height) to level 137 ($10\,\mathrm{m}$ above ground).

---

[1]Data is split based on distinct days. 20% of these distinct days are used for validation, while the remaining 80% serve as training dataset.

[2]See https://confluence.ecmwf.int/display/UDOC/L137+model+level+definitions.

## 2.3 Composition of Datasets

The two data sets are merged in order to obtain a tabular data shape. Each row of this tabular data refers to a spatio-temporal grid cell. Thus, it can be indexed by the longitude and latitude of its center as well as its hourly time stamp. Each row is either labeled as cell with lightning activity or without lightning activity. The nine ERA5 parameters (Table 1) on their 74 model levels enter the tabular data such that each resulting column refers to an *individual* parameter on an *individual* level, making up a total of $9 \cdot 74 = 666$ ERA5 feature columns. Further, each row is complemented with the information of the *hour of the day* and *day of the season* to account for diurnal and seasonal variations, respectively. Finally, the model topography[3] is added as another column.

## 3 Methods

To avoid incorporating expert knowledge by using specialized deep learning architectures and to efficiently handle a large number of input features, a classical fully connected neural network (Section 3.1) is used. To make sure that the neural network can model lightning sufficiently well and is worth being analyzed, the resulting outputs are compared to those of a state-of-the-art reference model (Section 3.2) on unseen test data. Finally, insights into the patterns exploited by the trained model are gained by applying Shapley additive explanations (Section 3.3).

## 3.1 Deep Learning Approach

A fully connected neural network was designed, consisting of eight hidden layers with $512 \times 512 \times 512 \times 512 \times 128 \times 128 \times 128 \times 16$ nodes. Leaky rectified linear unit (leaky ReLU) is used as activation function for all hidden layers. The input dimension is predetermined by the number of input features and thus equals $671$ (nine atmospheric variables on 74 levels, longitude, latitude, hour of the day, day of the season, and topography). The dimension of the output layer equals one, as it solely classifies whether the cell is with or without lightning activity. The model output is activated with the sigmoid function.

Prior to training, the input variables are standardized. For each of the atmospheric variables $v \in \{\mathrm{ciwc, clwc, crwc, cswc, q, t, u, v, w}\}$, the mean $\mu_v$ and standard deviation $\sigma_v$ are calculated over all 74 model levels together, but separately for each of the nine variables.

To prevent the model from overfitting, dropout (Srivastava et al., 2014) with a value of $0.15$ and early stopping with a patience of ten epochs are applied. Binary cross entropy serves as loss function with a weight of approximately $41$ for positive events (flash occurrences) to address for the highly imbalanced data set.

---

[3]The topography is represented by a single scalar value: the geopotential height from model level 137, which is the layer adjacent to the Earth's surface at the specified grid point.

**Table 2.** The reference model is trained using the following ten atmospheric variables.

| Description | Short Name |
|---|---|
| Convective available potential energy | `cape` |
| Binary indicator whether cloud is present | `cloud_exists` |
| Convective precipitation | `cp` |
| Mass of specific snow water content between the $-20°C$ and $-40°C$ isotherms | `cswc2040` |
| Cloud top height in height above ground | `cth` |
| Instantaneous surface sensible heat flux | `ishf` |
| Medium cloud cover | `mcc` |
| Total column supercooled liquid | `tcslw` |
| Mass of water vapor between the $-10°C$ and $-20°C$ isotherms | `wvc1020` |
| Two meter temperature | `2t` |

## 3.2 Reference Model

For reference a generalized additive model (GAM) (Wood, 2017) is used and fitted using an algorithm tailored for gigadata (Wood et al., 2017). This model is trained on longitude, latitude, hour of the day, day of the season, topography and the atmospheric variables listed in Table 2, which were derived from ERA5 soundings on meteorological expertise (Simon et al., 2023).

Thus, the input dimension for the reference model is only 15.

## 3.3 Explainability

While generalized additive models are interpretable by users (Lou et al., 2012), interpretability research of deep neural networks still suffers many gaps (Zhang et al., 2021). In this work SHAP (Lundberg and Lee, 2017) is utilized to gain insights into the patterns exploited by the neural network from Section 3.1 and to understand the features contributing to the classification of a spatio-temporal cell as one exhibiting lightning activity.

SHAP is a game theoretic approach which can be used to explain the relation of input and output of any machine learning model. It follows the concept of Shapley values (Shapley, 1952) to provide local interpretability by computing feature attributions that lead to the model's output for a given input. Unfortunately, the computation time for calculating exact Shapley values grows exponentially with the number of input features, leading to various ways in which Shapley values are operationalized (Sundararajan and Najmi, 2020; Chen et al., 2023). The two main approaches, observational and interventional, differ in the way they sample dropped input features to attribute for the difference between the model output and the expectation caused by the removed feature (Chen et al., 2020). While there is an ongoing debate about which approach is preferable (Chen et al., 2020), Janzing et al. (2020) argues, supported by experiments, that the observational approach is flawed and interventional provides the correct notion of dropping features.

This work applies Deep SHAP[4] (Lundberg and Lee, 2017) which is a model agnostic method that leverages extra knowledge about the nature of deep neural networks to approximate Shapley values more efficiently. The input features in this work are highly correlated, particularly along the vertical profiles within a single variable. Deep SHAP belongs to the family of interventional methods, thus effectively identifies the features that the model genuinely uses to generate a specific output, even in the presence of correlated inputs.

## 4 Results

This section first evaluates the performance of the deep learning approach and compares it to the reference model (Section 4.1). Next, the application of SHAP provides insights into the vertical profiles that the neural network found to be favorable for lightning (Section 4.2).

### 4.1 Performance of the deep learning approach

The neural network is trained as described in Section 3.1 to distinguish whether a given spatio-temporal cell is a cell with or without lightning activity. To map the model's output to a binary category, a threshold has to be defined. Due to the highly imbalanced nature of the given data set, this threshold is determined by maximizing the $F_1$ score, which balances precision and recall, on the validation set.

This study aims at finding the atmospheric patterns exploited by the neural network to classify cells being with or without
lightning, making the strategy and exact choice of threshold less critical. However, before analyzing the inner workings of the model it is essential to ensure that the trained model's performance is comparable to or even better than a state-of-the-art reference model.

The reference model is fitted as described in Section 3.2 and the threshold is computed following the same procedure.

From the confusion matrices displayed in Table 3 it can be concluded that the neural network slightly outperforms the
175 reference model in every category of the confusion matrix on previously unseen test data (year 2019). This is further supported by comparing the *Matthew correlation coefficients* (mcc) of the two models, where $+1$ represents a perfect match between model output and observations, and $0$ indicates no better than random guessing. The deep learning model has an mcc of approximately $0.278$, while the reference model has an mcc of $0.237$.

### 4.2 Identifying patterns exploited by the deep learning model

The performance of the deep learning approach encourages a closer examination of the patterns the model has learned to differentiate between cells with and without lightning activity. A sample is classified as having lightning activity when the model output exceeds the threshold $\phi$.

SHAP values (Section 3.3) indicate which inputs the neural network is particularly interested in. Given a specific input, the SHAP values of all input features always sum up, with only minor approximation errors, to the difference between a base value

---

[4]Provided by the `DeepExplainer` class within the Python package `shap`.

**Table 3.** Confusion matrices of the neural network model (left) and the reference model (right) on test year 2019.

|  |  | observed | | | | observed | |
|---|---|---|---|---|---|---|---|
|  |  | **yes** | **no** |  | | **yes** | **no** |
| **modeled** | **yes** | 14 372 | 61 431 | | **yes** | 12 654 | 65 176 |
|  | **no** | 15 766 | 1 374 756 | | **no** | 17 484 | 1 371 011 |

(derived from the expected model output based on so-called background data) and the actual model output. To identify patterns that are consistent across the entire training region and not influenced by the frequency of lightning in specific spatial cells, SHAP values and corresponding background data are calculated and sampled separately for each spatial cell. Specifically, for each spatial cell, the background data consists of the complete set of samples without lightning activity from that cell. To better understand the underlying patterns, the SHAP values are then scaled by dividing them by the difference between the base value

of the corresponding spatial cell and the threshold ($\phi$) at which a cell is classified as having lightning activity. This implies that the model classifies a sample to have lightning activity as soon as the scaled SHAP values sum up to one or more, regardless of the underlying base value and location.

Expressiveness is further improved by splitting the class of true positives into *less confident* and *very confident*. True positives with a model output in the interval $[\phi, \frac{1+\phi}{2})$ are considered *less confident true positives* and true positives with a model output

in $[\frac{1+\phi}{2}, 1]$ are termed *very confident true positives*.

The aggregated results of the scaled SHAP values of correctly classified cells with lightning activity are visualized in Fig. 1.

On average cloud ice (ciwc) and snow water content (cswc) contribute the most to the model's output. Also note that ciwc with its lighter-weighted ice crystals is particularly interesting at a geopotential height of approx. $8000$ to $12000\,\mathrm{m}$ and cswc with its solid precipitation at approx. $3000$ to $10000\,\mathrm{m}$.

Taking a closer look (Fig. 2) at the ciwc and cswc at these altitudes, it is noticeable that the model exhibits greater confidence when ciwc and cswc values are substantially elevated. Furthermore, there is a tendency for the model to produce false positives during periods of high ciwc and cswc, while false negatives are more prevalent when these values are low compared to correctly classified lightning events.

While classifications where a cloudy atmosphere is the most dominantly exploited feature by the neural network are the

205 majority, grouping the results into three categories, following Morgenstern et al. (2023), reveals additional patterns:

cloud: True positives where the sum of scaled SHAP values of ciwc, clwc, crwc and cswc over all model levels exceeds 0.5. Cloud-dominant cells with lightning activity are distributed across the entire region of interest, but are particularly abundant along the primary chain of the Alps.

mass: True positives where the sum of scaled SHAP values of q and t over all model levels exceeds 0.5. Mass-dominant cells

are predominantly situated in Northern Italy and Slovenia.

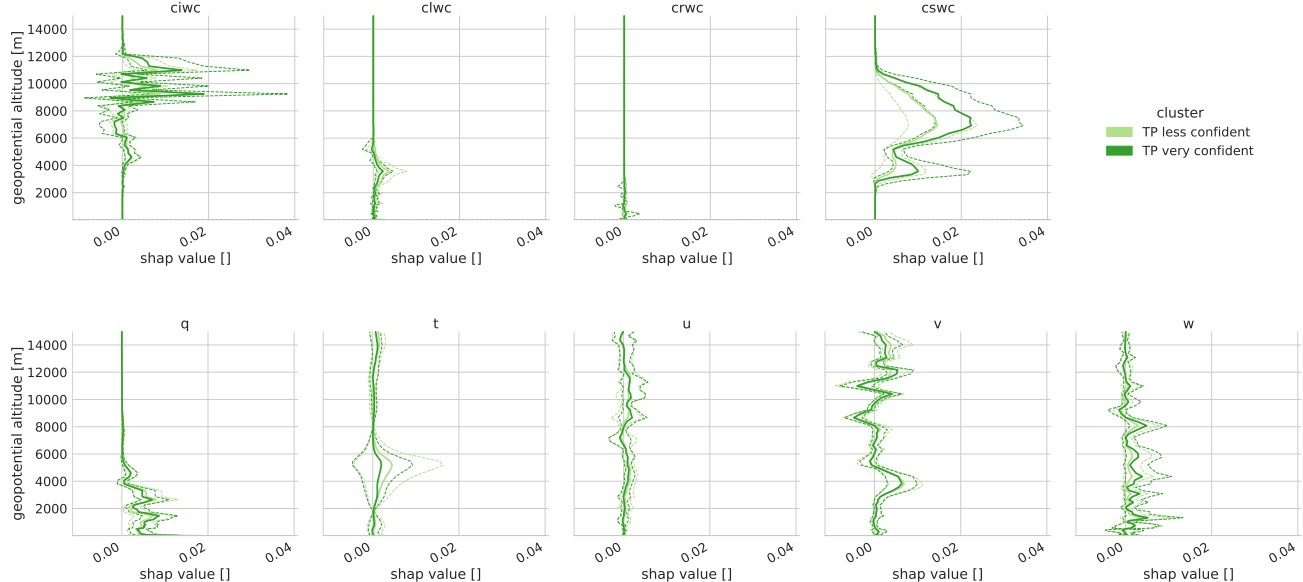

**Figure 1.** Scaled SHAP values for several variables (names on top of each subfigure) on correctly modeled lightning events (true positives). The two colors represent the confidence (stratified by median) of the network in its output. The dark green color summarizes the events where the network is very confident that a lightning event occurred. The light green color summarizes the events where the network still modeled correctly, but with less confidence. The solid lines show the median of all observations and the dashed lines highlight the interquartile range.

`wind`: True positives where the sum of scaled SHAP values of u, v and w over all model levels exceeds $0.5$. Wind-dominant cells are primarily concentrated in the northwestern region of the Italian flat terrain, the Po Plain.

Approximately 39.8% of the true positives belong to the cloud dominant, 2.6% to the mass dominant and 7.9% to the wind dominant class. Note that a single sample may belong to multiple groups or even none at all if the characteristics of cloud, mass, or wind are not distinctly pronounced.

Visualizing the vertical profiles of the real feature values (Fig. 3), their temperature profiles (t) are distinct. Events with high values for the mass-field have warmer temperatures and their temperatures decrease more strongly with height than the other two classes. This indicates that less work is required to displace particles in the vertical thus making it more prone to produce thunderstorm clouds. Since the maximum possible amount of water vapor in the air before condensation occurs is exponentially related to temperature via the Clausius-Clapeyron equation, events with high values of the mass-field also have by far the largest values for specific humidity q, particularly in the part of the atmosphere closest to the surface. When that water vapor condenses as air is lifted from near the surface the latent heat released during this phase change will heat the air and thus decrease its density and make a further rise of the air parcels more likely. Since there is so much more water vapor available for a phase change than with the other two categories, one would expect the category with high mass field values to also have higher amounts of liquid and solid water (ciwc, clwc, crwc, cswc) at altitudes above the level where the phase change occurs. However, the opposite is the case. The explanation rests in the difference of the horizontal size of a grid cell

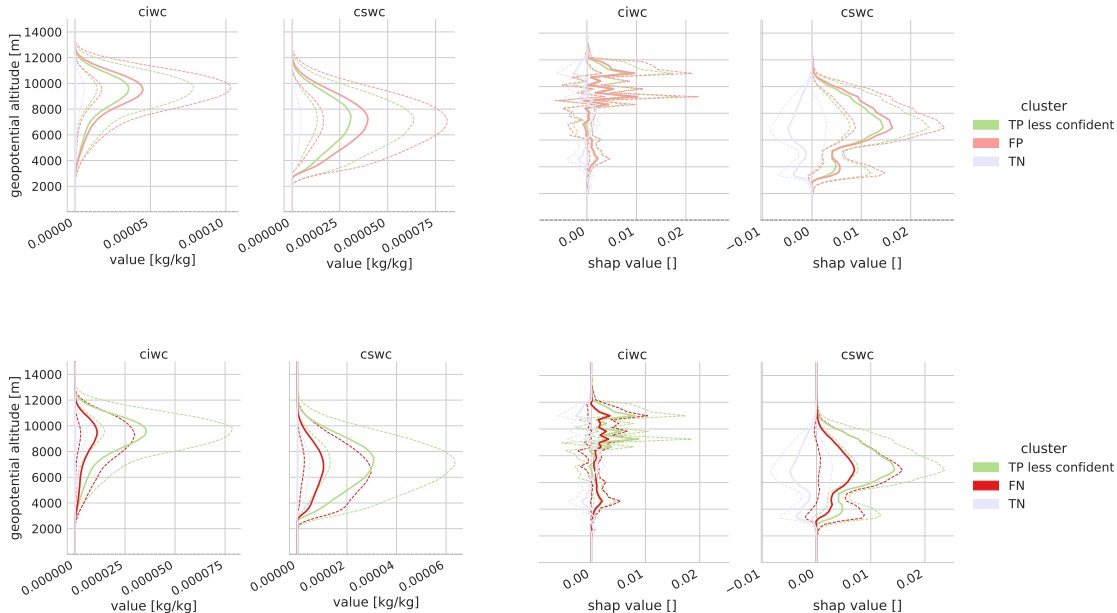

**Figure 2.** The two left columns display the vertical profiles of the real feature labels, while the two right columns present the vertical profiles of the scaled SHAP values. The upper row illustrates less confident true positives (TP) compared to false positives (FP), while the lower row illustrates less confident true positives compared to false negatives (FN). True negatives (TN) are also included for reference. The solid lines show the median of all observations and the dashed lines highlight the interquartile range.

of the ERA5 atmospheric reanalysis data, which is approximately $19\,\text{km} \times 28\,\text{km}$ in the region of interest, compared to the typical diameter of $5\,\text{km}$ of the most frequent type of thunderstorms - single cell thunderstorms (Markowski and Richardson, 2010). ERA5 data are average values over the whole grid cell and when only one single-cell thunderstorm occurs in an ERA5 grid cell, the average cloud-variables will be low since most of the ERA5 grid cell is cloud-free. The lowest absolute values of vertical velocity of all three categories support this conclusion. The deep learning approach thus has learned lightning from single cell thunderstorms.

The category with high wind-field values has the coldest temperature (t) profiles of all three categories and – because of the exponential relationship to maximum possible water vapor – also the lowest values of specific humidity (q) in the lower part of the atmosphere. Despite the least amount of water vapor available for condensation, this category has the largest amounts of cloud droplets (clwc) and of rain (crwc). Consequently such thunderstorms must occur in situations when most or all of an ERA5 grid cell is filled with clouds. Also, the absolute values of vertical velocity are largest of all three categories. The corresponding meteorological situations are large scale patterns of lifting in the atmosphere such as along cold fronts. Cold fronts in this region occur more frequently in the months between fall and spring, which explains why this category has the coldest temperatures. Also, cold fronts in this region typically occur in southwesterly flow downstream of the trough axis, which explains the exceptional large values of the v-component of the wind. Since wind speed also increases most strongly

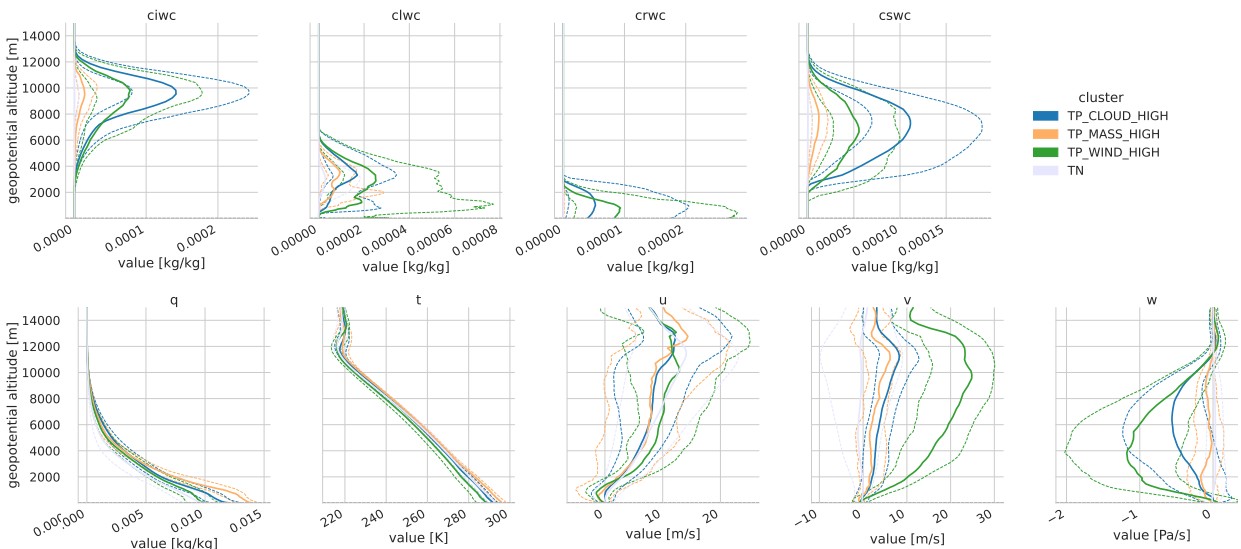

**Figure 3.** Vertical profiles of the real features per variable with colors indicating true negatives and different groups of true positives (cloud-, mass-, wind-dominant). The solid lines show the median of all observations and the dashed lines highlight the interquartile range. Note that in pressure coordinates, negative values of vertical velocity indicate upward motion.

with height, charge separation occurs on a tilted instead of a nearly vertical path as in mass-field lightning, having earned this type of lightning the name *tilted thunderstorm* (Brook et al., 1982; Takeuti et al., 1978; Takahashi et al., 2019; Wang et al., 2021).

The third category in Fig. 3 with high cloud-field variables has the largest amounts of solid water – ice crystals (ciwc), snow flakes and graupel (cswc) – but only the second largest amounts of liquid water (clwc, crwc). Also the vertical velocities are in between the other two categories. Therefore this category likely represents the meteorological situation of multicell and supercell thunderstorms (Markowski and Richardson, 2010), which have a larger footprint than single cell thunderstorms (the mass-field category) and will thus fill larger fractions of an ERA5 grid cell. This category could also contain cold front
situations (the wind-field category) where the cold front occupies only parts of an ERA5 grid cell.

        To test the hypothesis that the category with high cloud-field values contains both of these situations, i.e. mass-field and wind-field dominated situations, we divide this category into a cloud-mass and a cloud-wind category in Fig. 4. This is an approach also taken by Morgenstern et al. (2023). The grouping is based on whether the aggregate of scaled SHAP values is greater for mass-related or wind-related parameters.

And indeed, we find that the cloud-wind subcategory again has the largest amount of liquid water (clwc, crwc) and also larger values of the southerly wind component (v) indicative of the typical southwesterly flow for which (cold) fronts occur in this region. The cloud-wind category even has the higher solid water contents than the cloud-mass category indicating that even larger-sized thunderstorms in the absence of cold fronts do not always completely fill an ERA5 grid cell.

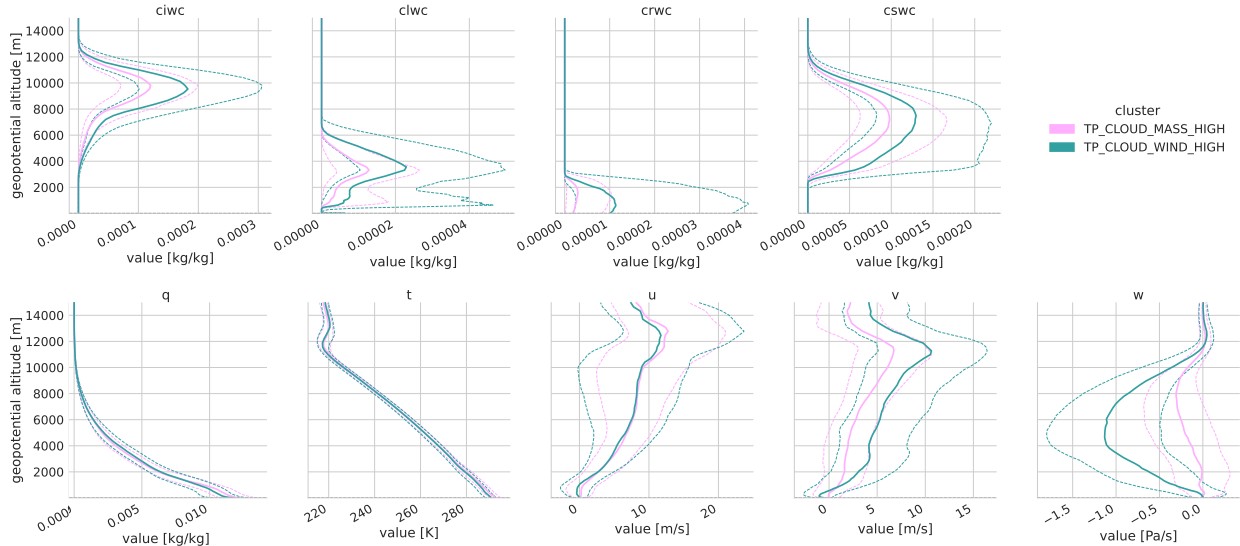

**Figure 4.** Vertical profiles of the real features per variable with colors indicating cloud-mass and cloud-wind dominant true positives. The solid lines show the median of all observations and the dashed lines highlight the interquartile range.

### 4.3 Sample case study

Thunderstorms and lightning commonly exhibit linear organization along meteorological boundaries such as fronts or convergence zones. Our deep learning model, trained exclusively on individual vertical atmospheric profiles, successfully identifies these linear structures without explicit knowledge of horizontal connections. A case study from June 20, 2019, demonstrates this capability. Two weak frontal systems occur in the region shown in Fig. 5. They are embedded within a region of high equivalent potential temperature (not shown). The bow-shaped front in the eastern half of the figure is more pronounced and extends over a larger part of the figure. The second one over Switzerland is only visible in the westernmost part of the figure. The deep learning approach model accurately reproduced the linear lightning pattern in the eastern region. However, it overestimated the width of the lightning zone and failed to capture its northernmost extent, as indicated by false positives (small green circles, Fig. 5). Nevertheless, the model exhibits deficiencies in reproducing the southwestern portion of the thunderstorm line over Switzerland, generating an erroneous linear feature further northward.

It is noteworthy that the threshold in this study was not chosen to perfectly calibrate the model, but instead to balance between precision and recall. Due to the heavy class imbalance, this generally results in overestimation.

## 5 Discussion and Conclusions

In this study, the region of interest are the Eastern Alps, a region that offers a variety of atmospheric processes due to its complex terrain and is well understood (Simon et al., 2023; Morgenstern et al., 2023). This is important, because it allows for

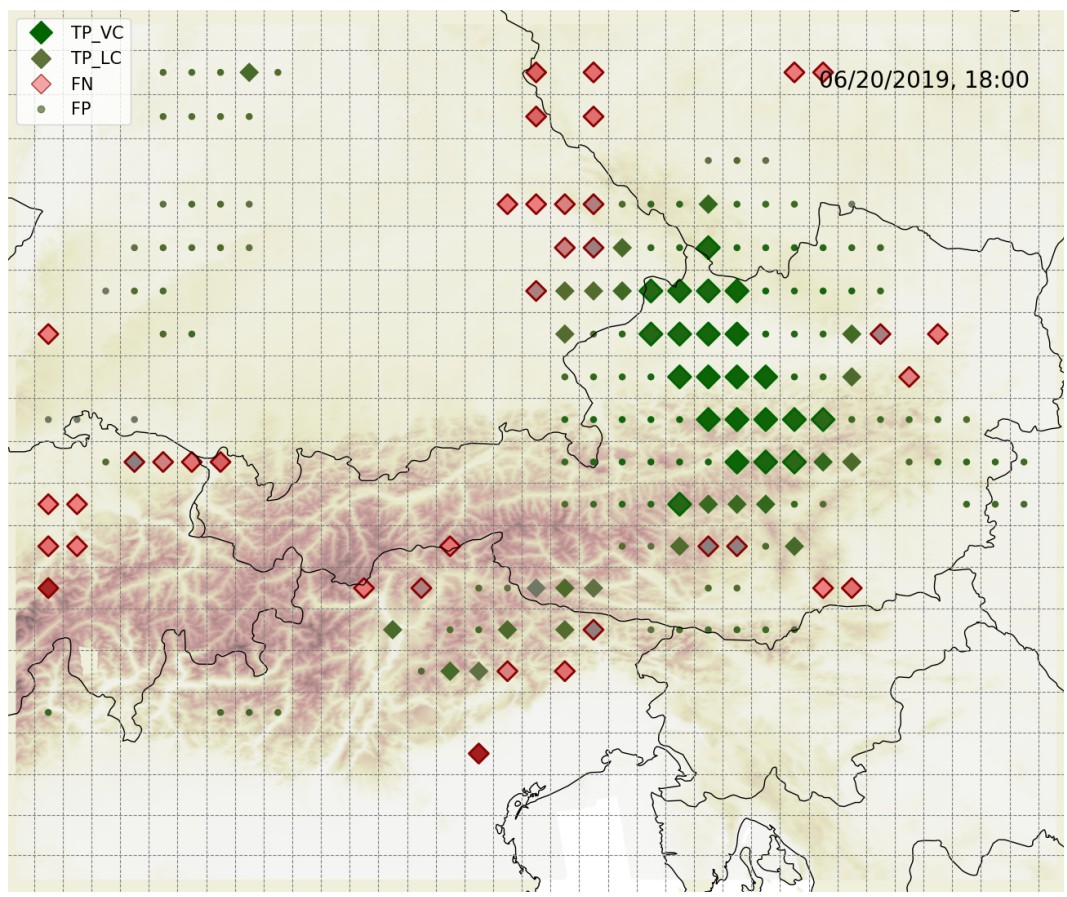

**Figure 5.** The map shows ERA5 grid cells with classifications of true positive (green diamonds), false negative (red diamonds) and false positive (dots) for the test data case June 20, 2019, in the hour before 18:00 UTC which is a case of the unseen test data. The size of the green diamonds indicates whether it is a *very* or *less* confident true positive. Low saturation of the red diamonds indicates that the output of the network was close to labeling the cell as one with lightning activity. The data for the displayed topography layer is taken from TanDEM-X (Rizzoli et al., 2017).

critical evaluation of the patterns uncovered by explainable AI methods and provides insights into whether this approach is suitable for accelerating scientific discovery in regions where knowledge is still scarce.

    A neural network is trained on the vertical columns of raw ERA5 data without inducing any further expert knowledge about atmospheric processes to classify whether there was a lightning event or not. Then scaled SHAP values are used to explain which variables and vertical levels attribute the most to correct classifications of cells with lightning activity. As indicated in
Section 4.2, the specific snow water and ice water content significantly capture attention, with peak interest occurring at a geopotential height of approximately $4000\,\mathrm{m}$ and $7000\,\mathrm{m}$ (cswc), and at heights of $9000\,\mathrm{m}$ and $11000\,\mathrm{m}$ (ciwc) respectively. Thus, the neural network discovered by itself the essential ingredient for lightning, namely charge separation. It occurs when ice crystals (ciwc) and larger frozen particles (graupel, cswc) are present in the convective updraft. Once the graupel is sufficiently heavy, its velocity is smaller than the velocity of the rising ice crystals, and the collisions between ice crystals and graupel
result in oppositely charged particles (Reynolds et al., 1957; Saunders et al., 2006). Lopez (2016, Fig. 1) shows the typical distribution of charges in a mature thunderstorm cloud. Additionally, it is noteworthy that the model seems to be particularly interested in the cloud ice water content at a height of $9000$ to $11000\,\mathrm{m}$ while recent literature usually examines the cloud ice water content at $440\,\mathrm{hPa}$ (typically about $6000\,\mathrm{m}$) (Finney et al., 2014, 2018; Silva et al., 2022). Focusing on the region between $9000$ and $11000\,\mathrm{m}$ means that it is crucial to vent ice particles all the way up to the tropopause and form anvils, as is
typical of thunderstorm clouds.

    Moreover, the model leverages the presence of southerly winds and vertical updrafts as reliable indicators for lightning occurrence especially in the northwestern Po Plain. Additionally, high specific humidity below $4000\,\mathrm{m}$ serves as a robust proxy in the central and eastern Po Plain, as well as in the southern regions of the Slovenian Alps.

    The case study in Section 4.3 demonstrates that, although recall and precision of the neural network may appear to be low at
first glance, the model effectively reproduces the general patterns of thunderstorms, despite overestimating and underestimating their extents. Similar observations were also made for many other examples not included in this manuscript.

    The results in this work suggest promising future applications. Being able to train a neural network directly on atmospheric soundings with good ability to distinguish between cells with and without lightning activity, and then opening the black box may enable researchers to gain a better understanding of atmospheric processes in regions like e.g. equatorial Africa where am-
ple studies are scarce (Chakraborty et al., 2022). The first MTG-I satellite was launched on 13 December 2022 and will provide a lightning imager (Holmlund et al., 2021) which appears to be a promising source for the target variable. Furthermore, many existing models come with two very different parameterizations for ocean and land (Finney et al., 2014) and this inevitably leads to discontinuities in coastal areas. Also the reasons for the much lower lightning frequency over ocean are not as well understood yet. XAI might be a valuable building block in moving towards a more holistic understanding of the underlying
atmospheric processes.

    Using ML models to find parametrizations require them to be generalizable. In Ehrensperger et al. (2023) a similar model was trained on the same region but without using longitude, latitude and the day of the year as input features. While not giving the location to the model still provided a comparable performance, it enabled to evaluate the model on Continental Europe.

The results show that the model is still able to perform comparably well on landcovered areas on previously unseen test data, demonstrating its ability to generalize across both time and location.

Future work might improve the results presented in this study. Here, a simple fully connected neural network is used and therefore the model loses information about the connectivity of the values along the levels of the vertical profiles. Using convolutional layers to process the profiles would, most likely, improve the results.

Convection and cloud processes are not purely vertical processes and thus ML parameterization greatly benefits from using multiple neighboring vertical atmospheric columns instead of a single column. Wang et al. (2022) work with $192\,\text{km} \times 192\,\text{km}$ grid cells to model, among others, subgrid zonal and meridional momentum flux due to vertical advection and suggest that a $3 \times 3$ subgrid could further improve the performance of the deep learning approach.

*Code and data availability.* The software (version 1.2; Python and R code) used to produce the results and plots in this manuscript is licenced under MIT and published on Zenodo (https://dx.doi.org/10.5281/zenodo.13907708) (Ehrensperger et al., 2024). The source code relies on two data sources:

1. ERA5 (Hersbach et al., 2020) data are available via the Climate Data Store (Hersbach et al., 2018, 2017). Scripts for sending the retrievals are included in the `data-preprocessing` directory of the Zenodo repository (Ehrensperger et al., 2024).

2. ALDIS (Schulz et al., 2016) data was aggregated to align with the spatio-temporal grid cells of ERA5 for use in this work. The transformed data is published in Simon et al. (2024).

*Author contributions.* **Gregor Ehrensperger**: Methodology, Software - model & explainable AI & plotting & data preparation, Writing – original draft. **Thorsten Simon**: Data curation, Software - reference model & plotting, Writing – original draft. **Georg Mayr**: Supervision, Writing - review & editing. **Tobias Hell**: Conceptualization, Methodology.

*Competing interests.* The authors declare that they have no known competing financial interests or personal relationships that could have appeared to influence the work reported in this paper.

*Acknowledgements.* We are grateful for data support provided by Gerhard Diendorfer and Wolfgang Schulz from OVE-ALDIS. We also thank Deborah Morgenstern and Johannes Horak for their script to compute geopotential height on ERA5 model levels. Additionally, we thank Johanna Rissbacher for contributing parts of Fig. 5 and the corresponding code. Furthermore, we greatly appreciate the insightful and constructive reviews from the anonymous reviewers, which have significantly enhanced the quality of this paper.

*Financial support.* This work was funded by the Austrian Science Fund (FWF, grant no. P 31836) and the Austrian Research Promotion Agency (FFG, grant no. 872656).

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
