# Peer review of "Identifying Lightning Processes in ERA5 Soundings with Deep Learning"

_EGUsphere, 2024_

## Referee Comment (RC1)

**Review of egusphere-2024-1718 : "Identifying lightning processes in ERA5 soundings with deep learning"**

This manuscript presents a simple deep neural network model which predicts lightning location based on vertical profiles from ERA5 reanalysis data and evaluates its performance compared to a refence model (GAM – generalised additive model). SHAP values are then used to identify which particular features of the input profiles are important in determining the occurrence of lightning. Since the physical processes which generate lightning are not explicitly represented in weather or climate models

The work is a nice demonstration of the use of machine learning for both prediction and process understanding, although the motivation of the paper and the novelty of the findings is not well highlighted in the current manuscript. There are also a number of aspects which need clearer description and some of the results either require more in depth analysis, or just removing. Overall, the manuscript needs revision before it could be considered for publication. I include more specific comments below.

Major comments:

1) The rationale for the work is not completely clear, in particular I think you could do a better job of identifying the novelty of the research compared to previous studies. What new do you learn here compared to previous studies by the co-authors? Or is the novelty really in the method? Is it better (figure 1 / table 3 suggests it is slightly better in some cases, but not hugely – it still misses > half of cases of lightning, but no lightning was seen in about 80% of the modelled "yes" cases). I would be clear about what you are aiming to do up front. I wasn't clear what you meant by a "holistic description of lightning" (top of p3). The physical insight offered by explainable AI could be really interesting, but I didn't feel you really took this very far in terms of discussing your results, certainly not much beyond confirming what we already know.

2) A number of decisions are made (e.g. choice of ML method, choice of model variables to include) without a clear justification. I think the variables you have chosen are useful, but there are other things you have neglected (e.g. the height / pressure on the model levels and surface fields) which could be relevant. In particular, as I understand it you have chosen a method which does not know about the links between adjacent levels. Does this mean the model does not know about derivatives? This could be particularly important when thinking about things like stability. I wasn't quite clear what "topography" means. Is this just the height of the grid cell? What about some measure of slope / sub grid variability? Would this not be relevant too for convection?

3) Another example of choices is the various parameters related to the training (e.g. dropout, early stopping patience) which are just given without justification.

4) I appreciate that there is a formal requirement for inputs to be independent for SHAP values to be calculated using Deep SHAP. Since this is clearly not the case here, I think you need to be more careful in explaining why it is ok to use this algorithm, but also more fundamentally explain what the SHAP values will tell you when the variables are highly correlated (as adjacent points in a profile are likely to be). Looking at some of your results they are very noisy (e.g. CIWC in figure 2). Is this "real" or is this a feature of the implementation of SHAP you are using?

5) Training / validating / test data sets. As I read the paper you use 2010-2018 for training the model and for validating the model (tuning and preventing overfitting). How do you split this data between training and validating? Then 2019 is reserved as a truly independent test data set for evaluating the overall model. Please be clear on this at the start.

6) Subdomains: having described the training of the model and the calculation of an appropriate threshold for producing a binary lighting / no lightning output, you then go on to say that you subdivide the data into four subdomains. There is no clear description of where these subdomains are. If you use them then I think you need to say how they are defined / include a map. Do you retune the output threshold for each subdomain separately? I am not quite clear from the manuscript. If so, why is this necessary and what difference does it make? It would seem to limit the universality of the method if you need a different threshold for different regions. It also makes the precise choice of region quite important (and another arbitrary decision).

7) Actually, re-reading you say "In this case, the model's threshold is calibrated to align the average predicted and observed lightning frequencies of the validation set". This is very different to what you did on the previous page and makes the results of figure 1 look much better. This is very misleading. Please explain clearly why you need to calculate thresholds in two different ways and avoid doing so if it is at all possible.

8) How do you calculate model confidence? A number of figures show categories split into less confident / very confident and it is mentioned in the text, but you don't actually explain how you calculate this? Is it also based on the threshold?

9) On p8 you categorise the data based on the most important group of features (cloud, mass, wind). Identification of a lightning prediction only requires the sum of the scaled SHAP values to exceed 1, so in theory all three subgroups could exceed a summed SHAP value of >0.5. What do you do in that case? What about the case where all 3 groups are less than <0.5 but sum to >1.0? What are the relative frequencies of occurrence of these different categories? Why later do you go and split cloud into cloud-wind and cloud-mass?

10) Figures and use of colour. I found it hard to read many of the figures. E.g. in figure 2 I could not see the pale green mean line for "TP less confident". I also really struggled to see the boundary where pale and dark green shading overlapped. Please consider whether it might be better to use e.g. dashed lines to mark the boundaries of the shaded regions. Similarly in figures 3-5. Also consider the choice of colours in figures 3-5. For those who are colour blind it would be impossible to distinguish these overlapping colours. Red/green is a particularly bad combination for many people.

11) Figure 6 – spatial distribution by category. I wasn't quite sure what the take-home message here was. There is only a very short paragraph which describes the figure but does not really discuss the results at all. Either this needs more analysis, or if it doesn't add anything just remove the figure.

12) Case study (figure 7). Again, I wasn't quite sure what I was supposed to take away from looking at a single case study. The model gets some bits right but tends to predict lightning over too wide an area. Without any meteorological context it is hard to know why. Is this a general result? This case study either needs better justification and more discussion if there is an important result to take from it, or otherwise just remove it.

13) In the introduction you mentioned the potential for using ML models for parametrisation. This would require the model to be generalisable. I wonder if you can revisit this in the discussion and conclusions. How widely applicable is your model? It seems to respond to different features in different regions. Have you tried applying to other unseen regions /

seasons? What would you need to do to make it more generalisable? At least this is worth discussing.

Minor comments:

1) Abstract, line 2. "wind shears" -> "wind shear"
2) Abstract, line 13. "as physically meaningful" -> "as a physically meaningful"
3) p2, lines 4-5. The phrase "numerical computations" sounds odd. How about something like "The term *proxy* is commonly used for quantities derived from model output *after* the simulations has run. *Parametrizations* diagnose lightning *while* the model is running and hence can feed back on the simulation."
4) p2, line 8. "perform reasonably good" -> "perform reasonably well"
5) p3, 3rd paragraph. I found the sentence ending with "(Sect 3)" a bit confusing. It is ambiguous what "(Sect 3)" refers to. Perhaps delete and then change the following sentence to read "Section 3 describes the two modelling approaches and additionally illustrates …" This makes it clearer what each section is about.
6) Table 1. Units should be in roman not italic font by convention.
7) p5, section 3.1, lines 6-7. ".. are standardized by considering the 74 levels altogether, prior training". I do not understand what this means. Please explain.
8) P6, line 1. I guess you are using the implementation of DeepSHAP from Lunberg available on GitHub? If so, perhaps say so explicitly / provide a reference. The footnote on p8 mentions *DeepExplainer*.
9) Figure 2 caption "The coloured areas highlighted the 50% quantiles." This is a bit unclear. Do you mean the shaded are shows the interquartile ranges (25%-75%)? If so, is this of the mean SHAP values calculated at each cell point, or is it the interquartile range of all SHAP values across all cells?
10) p13, line 6. "cyrstals" -> "crystals"
11) p15, line 12. "MGT-I" -> "MTG-I"

---

## Author Response (AR1)

We sincerely appreciate the time and effort the editor and reviewers have dedicated to evaluating our manuscript and are grateful for the valuable feedback, which has significantly enhanced the readability and clarity of our work. We have carefully considered all the feedback and made the necessary revisions. Below, we provide detailed, point-by-point responses to the chief editor and each of the reviewers' comments, along with any additional changes made to the manuscript. The last Section lists additional changes which were made while proofreading the manuscript.

**Please note that the line numbers and figure numbering mentioned in our responses correspond to the file of the author's tracked changes (diff between submitted preprint and first revision).**

**Reply to chief editor Dr. Añel (CEC1, CEC2)**

**CEC1**

> After checking your manuscript, we have found several issues regarding compliance with our journal's Code and Data Availability policy. You have done good work regarding sharing the code. The only problem here is that, in the corresponding section of the manuscript, you link a GitHub repository. GitHub is not suitable for long-term archival of assets for scientific research. GitHub itself states it. You have a Zenodo repository (http://dx.doi.org/10.5281/zenodo.10899180). You must cite this Zenodo repository instead of GitHub in the Code Availability section. Please, note this for potentially reviewed versions of your manuscript.
>
> We are more concerned that you have not shared the ALDI data or the merged dataset you use. We must clarify if you qualify for an exception to our policy regarding sharing the data. In this way, we would expect that you had shared the merged dataset that you mention in the manuscript; as it is not the original ALDI data, I understand that this should be possible. Anyway, we need some additional evidence or justification about why the ALDI data can not be shared: laws, regulations, a license, etc.
>
> Finally, a minor issue: note that in your instructions for using the model, you provide indications on the dependency on Openjdk8, and for installation, you provide a command that is only for Debian-based operative systems. Although obvious, you could want to modify it by simply mentioning the dependency.

**CEC2**

Regarding the ALDI data, you need more than just referring readers to a third-party webpage. You use this dataset as an integral part of your work, and therefore, we need it to be published. Moreover, you state that you actually use a new merged dataset. At a minimum, you have to publish the new merged dataset (which I understand you own) in an acceptable public repository according to our policy.

It is imperative that you take the initiative to coordinate with the owners of the ALDI dataset for its publication in a new repository. This is particularly important as you intend to use it for the publication of your submitted manuscript. It is not acceptable for readers wishing to replicate your work to point them out to an unreliable webpage that does not have the data and for which they could get access denied. The data must be public and in trustable repositories to avoid precisely these issues.

Also, given that you do not use the ALDI dataset directly in your work, it would be good if you discussed with the distributors of the ALDI dataset its publication in an acceptable repository for scientific publication.

Therefore, it is a condition for the acceptance of your manuscript for publication in GMD that you publish the new merged dataset that you use in your work. I must emphasize that failure to do so will result in the rejection of your manuscript.

Thanks for insisting on making our research easier to reproduce!

- The transformed ALDIS data is available at [1].

- The READMEs included with the source code [2] accompanying our paper have been updated accordingly.

- The revised version of the paper refers to [1] for obtaining the transformed ALDIS data set (lines 415–417).

- GitHub-Links to our source code are replaced by links to the Zenodo-repository. During work on our revised manuscript, we also updated the source code from 1.1 to 1.2. See lines 410–411 and 414.

[1] https://doi.org/10.5281/zenodo.13164463

[2] https://dx.doi.org/10.5281/zenodo.13907708

**Reply to anonymous Referee 1 (RC1)**

This manuscript presents a simple deep neural network model which predicts lightning location based on vertical profiles from ERA5 reanalysis data and evaluates its performance compared to a refence model (GAM – generalised additive model). SHAP values are then used to identify which particular features of the input profiles are important in determining the occurrence of lightning. Since the physical processes which generate lightning are not explicitly represented in weather or climate models The work is a nice demonstration of the use of machine learning for both prediction and process understanding, although the motivation of the paper and the novelty of the findings is not well highlighted in the current manuscript. There are also a number of aspects which need clearer description and some of the results either require more in depth analysis, or just removing. Overall, the manuscript needs revision before it could be considered for publication. I include more specific comments below.

We appreciate and want to thank for your exceptionally careful and detailed review. We have revised the manuscript based on your comments and suggestions and are confident that it has significantly improved in clarity and overall quality.

**Major comments**

**MajC1**

> The rationale for the work is not completely clear, in particular I think you could do a better job of identifying the novelty of the research compared to previous studies. What new do you learn here compared to previous studies by the co-authors? Or is the novelty really in the method? Is it better (figure 1 / table 3 suggests it is slightly better in some cases, but not hugely – it still misses > half of cases of lightning, but no lightning was seen in about 80% of the modelled "yes" cases). I would be clear about what you are aiming to do up front. I wasn't clear what you meant by a "holistic description of lightning" (top of p3). The physical insight offered by explainable AI could be really interesting, but I didn't feel you really took this very far in terms of discussing your results, certainly not much beyond confirming what we already know.

Thank you for highlighting the need to emphasize the novelty and our objectives more clearly.

We have thoroughly revised the abstract (lines 3–19, 22–26), Section 1 (Introduction; lines 61–62, 73–109), Section 3 (Methods; line 145) Section 4.1 (Performance of the Deep Learning Approach; lines 203–208), Section 4.2 (Identifying patterns exploited by the deep learning model; lines 237-239), Section 5 (Discussion and Conclusions; lines 359–362, 397–399, 403–409) to enhance clarity.

In summary, this study aims at finding the atmospheric patterns exploited by the neural network to classify cells being with or without lightning, making the strategy and exact choice of threshold (which balances recall and precision) less critical. However, before analyzing the inner workings of the model it is essential to ensure that the trained model's performance is comparable to or even better than a state-of-the-art reference model. Our work directly utilizes raw model-level data instead of relying on expert-selected variables derived from vertical profiles. This on one hand makes the methodology easier transferable to other regions with less expert knowledge and on the other hand showcases the capability of AI to extract meaningful patterns. Due to the high number of correlated input features, commonly used plots for visualizing SHAP values are not feasible for interpretation. Therefore, we aggregated the results for a more global understanding, introduced scaled SHAP values for improved explainability, and visualized the median and quartiles of the results as vertical profiles to aid interpretation.

**MajC2**

> A number of decisions are made (e.g. choice of ML method, choice of model variables to include) without a clear justification. I think the variables you have chosen are useful, but there are other things you have neglected (e.g. the height / pressure on the model levels and surface fields) which could be relevant. In particular, as I understand it you have chosen a method which does not know about the links between adjacent levels. Does this mean the model does not know about derivatives? This could be particularly important when thinking about things like stability. I wasn't quite clear what "topography" means. Is this just the height of the grid cell? What about some measure of slope / sub grid variability? Would this not be relevant too for convection?

Thanks for pointing out that we missed to explain the topography variable. Like you assume, it is the geopotential height at model level 137 (adjacent to surface) of the given grid cell. We have added a footnote (line 140). The choice for the Neural Network was made (among other reasons) because it is able to handle a large number of input features, is comparably fast to train and performs well in many complex classification tasks. We included a note about the number of input features in the first paragraph of Section 3 (Methods; lines 142–144).

We did not include surface fields, since we solely work with "raw" model level data. This sets our study apart from other studies. Pressure on the model level would indeed be an interesting additional variable, which we will consider for future research. In this study, we had to keep the number of overall parameters low to ensure that computing time and memory usage remain managable.

While slope, sub grid variability and similar properties would be important for physical modelling, the neural network will basically learn to "fingerprint". We provided longitude and latitude as input, thus the model is able to grasp that different "atmospheric patterns" are important at different locations (aka sub-grid topography). Also, since we are using a fully connected neural network which basically allows for all linear combinations of any inputs, the model is able to access the derivatives by learning the parameters (weights and biases) accordingly.

**MajC3**

> Another example of choices is the various parameters related to the training (e.g. dropout, early stopping patience) which are just given without justification.

The justification for adding dropout and early stopping patience is to prevent the model from overfitting. We decided not to add more description regarding the concrete choices of parameters, as these are commonly known best practices in the machine learning community, and the specific parameter values are within the commonly used ranges.

However, we would like to provide a more detailed explanation here: - Hidden nodes per layer: The first hidden layers approximately match the size of the input layer to capture the full complexity initially. The last hidden layers are smaller to save computational power.

- Number of layers: Sufficiently deep to fit more complex patterns.

- Leaky ReLU: Computationally fast; the gradient does not vanish with negative input.

- Sigmoid function on output layer: Ensures the output is between 0 and 1 and is commonly used for binary classification.

- Mean-standard scaling: Input needs scaling. We have experimented with mean-standard and min-max scaling; the former performed notably better.

- Dropout: Used to prevent overfitting, especially with highly imbalanced data. We experimented with values between 0.1 and 0.6. The specific value did not significantly affect performance, but the model benefited from using dropout.

- Early stopping patience: Due to the vast amount of data, the model converges within a few epochs. Each epoch took around one hour. Early stopping ensures that training continues as long as performance improves and stops once it plateaus, preventing overfitting.

- Binary cross-entropy loss function: Commonly used for binary classification problems. Weighting positive events proportionally to their relative occurrence is best practice for imbalanced classification tasks.

While these being the final parameter choices, we of course experimented with number of layers, number of nodes, different activation functions and other parameters to ensure that the model's training is stable and performs well.

**MajC4**

> I appreciate that there is a formal requirement for inputs to be independent for SHAP values to be calculated using Deep SHAP. Since this is clearly not the case here, I think you need to be more careful in explaining why it is ok to use this algorithm, but also more fundamentally explain what the SHAP values will tell you when the variables are highly correlated (as adjacent points in a profile are likely to be). Looking at some of your results they are very noisy (e.g. CIWC in figure 2). Is this "real" or is this a feature of the implementation of SHAP you are using?

The assumption that inputs are independent, the consequences when this condition is not met and which approach for sampling Shapley values should be the preferred one, are sources of many debates in the literature [1, 2].

Scott Lundberg (creator of the SHAP library) writes in a discussion [3]:

> The original SHAP paper proposed pure conditional expectations for measuring the value of a set of input features, and then proposed using the Shapley values to reduce this exponential number of values down to a single number for each feature. To make things more tractable we can assume feature independence. This is of course never true in practice, and so may seem like a terrible approximation. But it turns out that you can look at this assumption from a very different perspective, where you break feature dependence not because of an independence assumption, but because of arguments based on causal inference.

To summarize: There are two main approaches to approximate Shapley values which fundamentally differ in the way they sample left-out (dropped) features to account for feature attribution. The interventional approach (used by Deep SHAP, which we employed in our work) treats inputs as independent and thereby identifying which inputs are genuinely used by the model. In the case of correlated inputs the trained model might not give equal contribution to all correlated variables and since the interventional Shapley values are "true to the model", also the Shapley values will only reflect the input variables the model actually uses. In the context of our work, we believe that using Deep SHAP is the right choice. We agree that the current paragraph in the manuscript is rather confusing and thus will replace the second and third paragraph of Section 3.3 (Explainability) with a clearer argumentation. Following up on the previous explanations, the noise observed in CIWC in Figure 1 is indeed real to the model. This figure indicates that the model relies more on the model levels of CIWC where SHAP peaks compared to model levels with no or lower peaks.

We have added more clarity and a more detailed description by updating the relevant parts in Section 3.3 (Explainability; lines 176–182, 184–194).

[1] Chen, Hugh, et al. "True to the model or true to the data?." arXiv preprint arXiv:2006.16234 (2020).

[2] Janzing, Dominik, Lenon Minorics, and Patrick Blöbaum. "Feature relevance quantification in explainable AI: A causal problem." International Conference on artificial intelligence and statistics. PMLR, 2020.

[3] christophM. discussion. GitHub. https://github.com/christophM/interpretable-ml-book/issues/142#issuecomment-564681746

**MajC5**

> Training / validating / test data sets. As I read the paper you use 2010-2018 for training the model and for validating the model (tuning and preventing overfitting). How do you split this data between training and validating? Then 2019 is reserved as a truly independent test data set for evaluating the overall model. Please be clear on this at the start.

Data is split based on distinct days. 20% of these distinct days are used for validation, while the remaining 80% serve as training dataset. We added this information in the revised version in Section 2 (Data; lines 115–116).

**MajC6**

> Subdomains: having described the training of the model and the calculation of an appropriate threshold for producing a binary lighting / no lightning output, you then go on to say that you subdivide the data into four subdomains. There is no clear description of where these subdomains are. If you use them then I think you need to say how they are defined / include a map. Do you retune the output threshold for each subdomain separately? I am not quite clear from the manuscript. If so, why is this necessary and what difference does it make? It would seem to limit the universality of the method if you need a different threshold for different regions. It also makes the precise choice of region quite important (and another arbitrary decision).

Division in subdomains was only needed to illustrate the effect of two different thresholding methods (based on F1 score or calibrated on the actual number of lightning events) on the diurnal cycle of lightning. These subdomains had been chosen for differences in the diurnal cycle due to topographic differences between them. Since we have deleted the comparison in Section 4.1 (Performance of the deep learning approach; 215–235) between the two methods as too much of a tangent topic in response to your questioning its usefulness, subdomains are no longer needed. We want to note that we did not retune the output threshold based on subdomains or locations, as doing so would limit the universality of this method, as you pointed out.

**MajC7**

> Actually, re-reading you say "In this case, the model's threshold is calibrated to align the average predicted and observed lightning frequencies of the validation set". This is very different to what you did on the previous page and makes the results of figure 1 look much better. This is very misleading. Please explain clearly why you need to calculate thresholds in two different ways and avoid doing so if it is at all possible.

We acknowledge that introducing a second threshold to discuss a tangent topic is rather confusing. As mentioned in our response to point 6, we have removed this comparison altogether.

**MajC8**

> How do you calculate model confidence? A number of figures show categories split into less confident / very confident and it is mentioned in the text, but you don't actually explain how you calculate this? Is it also based on the threshold?

Like you assume, true positives are split into "less confident" and "very confident" based on threshold phi. True positives are given by model outputs greater than phi. The true positive category is further subdivided into less and very confident true positives. Less confident true positives are given by correctly classified samples with a model output smaller than (1 + phi) / 2 and very confident true positives by outputs greater than or equal to (1 + phi) / 2. We have added this information in Section 4.2 (Identifying patterns exploited by the deep learning model; lines 249, 252–254) of the revised manuscript.

**MajC9**

On p8 you categorise the data based on the most important group of features (cloud, mass, wind). Identification of a lightning prediction only requires the sum of the scaled SHAP values to exceed 1, so in theory all three subgroups could exceed a summed SHAP value of >0.5. What do you do in that case? What about the case where all 3 groups are less than <0.5 but sum to >1.0? What are the relative frequencies of occurrence of these different categories? Why later do you go and split cloud into cloud-wind and cloud-mass?

A single sample can be attributed to one, multiple or even none of the three categories. Approximately 39.8% of the true positives belong to the cloud-dominant, 2.6% to the mass-dominant and 7.9% to the wind-dominant class. We have supplemented the manuscript (Section 4.2 (Identifying patterns exploited by the deep learning model; lines 273–275)) with this information. A more comprehensive list is provided here but not added to the paper to avoid bloating the manuscript:

- Number of samples in cloud-dominant TPs: 5726 (39.8% of TPs)
- Number of samples in mass-dominant TPs: 380 (2.6% of TPs)
- Number of samples in wind-dominant TPs: 1133 (7.9% of TPs)
- Number of samples TPs without dominance: 7331 (51% of TPs)
- Number of samples being cloud and mass dominant at the same time: 7
- Number of samples being cloud and wind dominant at the same time: 191
- Number of samples being cloud, wind, and mass dominant at the same time: 0
- Number of samples being mass and wind dominant at the same time: 0

The motivation to split the cloud-dominant class into cloud-mass and cloud-wind is made more clearly in Section 4.2 (Identifying patterns exploited by the deep learning model; lines 317–325).

**MajC10**

Figures and use of colour. I found it hard to read many of the figures. E.g. in figure 2 I could not see the pale green mean line for "TP less confident". I also really struggled to see the boundary where pale and dark green shading overlapped. Please consider whether it might be better to use e.g. dashed lines to mark the boundaries of the shaded regions. Similarly in figures 3-5. Also consider the choice of colours in figures 3-5. For those who are colour blind it would be impossible to distinguish these overlapping colours. Red/green is a particularly bad combination for many people.

Thank you for suggesting the use of dashed lines to mark the boundaries. We have adapted Figs. 1, 2, 3, and 4 accordingly. Additionally, we removed the shaded areas as they bloated the graphics and slightly affected the colors due to the transparency settings. We also removed Figs. 4a and 5a (old numbering), since they were quite hard to read and also did not add a lot of value. We used http://hclwizard.org:3000/cvdemulator/ to check on the colors and they should be suitable for color-blind persons.

**MajC11**

Figure 6 – spatial distribution by category. I wasn't quite sure what the take-home message here was. There is only a very short paragraph which describes the figure but does not really discuss the results at all. Either this needs more analysis, or if it doesn't add anything just remove the figure.

The idea was to demonstrate the model's ability to pick up on regional differences, but we agree that the figure itself does not add a lot of value. Thus we removed Figure 6 in the revised version and added the content in text form to Section 4.2

(Identifying patterns exploited by the deep learning model; lines 267–272).

**MajC12**

> Case study (figure 7). Again, I wasn't quite sure what I was supposed to take away from looking at a single case study. The model gets some bits right but tends to predict lightning over too wide an area. Without any meteorological context it is hard to know why. Is this a general result? This case study either needs better justification and more discussion if there is an important result to take from it, or otherwise just remove it.

We have added a more detailed explanation and thoroughly revised Section 4.3 (Sample case study; lines 351–357), and added paragraph 4 in Section 5 (Discussion and Conclusions; lines 380–382).

The reason why we added the case study is because researchers interested in applying the method are usually also interested in seeing concrete examples / case studies. If desired we could also move this part into the Appendix.

**MajC13**

> In the introduction you mentioned the potential for using ML models for parametrisation. This would require the model to be generalisable. I wonder if you can revisit this in the discussion and conclusions. How widely applicable is your model? It seems to respond to different features in different regions. Have you tried applying to other unseen regions / seasons? What would you need to do to make it more generalisable? At least this is worth discussing.

This is a very interesting thought. We actually performed experiments on generalisation and presented the results in a poster session at the EGU 23 [4] (the poster itself is downloadable as supplement material). To summarize: We have trained a model on the same area using the summer months, but without longitude/latitude and without the day of the year. We then used this model to classify cells with lightning activity on a much bigger area (Continental Europe). The model clearly learned the patterns on landcovered areas, but like we also observe in the case study, overestimates lightning activity. This is mainly due to the choice of threshold and the difficulty of accurately finding the spatial and temporal extend of the lightning event. We have also added this discussion to the manuscript in Section 5 (Discussion and Conclusions; lines 392–396).

[4] G. Ehrensperger, T. Hell, G. J. Mayr, and T. Simon, "Evaluating the generalization ability of a deep learning model trained to detect cloud-to-ground lightning on raw ERA5 data," Copernicus Meetings, Feb. 2023. doi: 10.5194/egusphere-egu23-15817.

**Minor comments**

**MinC1, MinC2, MinC4, MinC10, MinC11**

- 1. Abstract, line 2. "wind shears" -> "wind shear"

- 2. Abstract, line 13. "as physically meaningful" -> "as a physically meaningful"

- 4. p2, line 8. "perform reasonably good" -> "perform reasonably well"

- 10. p13, line 6. "cyrstals" -> "crystals"

- 11. p15, line 12. "MGT-I" -> "MTG-I"

Thanks for the list of typos. We have corrected the manuscript accordingly:

- 1: line 2

- 2: line 21

- 4: line 43

- 10: line 369

- 11: line 386

**MinC3**

> p2, lines 4-5. The phrase "numerical computations" sounds odd. How about something like "The term proxy is commonly used for quantities derived from model output after the simulations has run. Parametrizations diagnose lightning while the model is running and hence can feed back on the simulation."

Thanks. We followed your suggestion, see lines 38–40 in Section 1 (Introduction).

**MinC5**

> p3, 3rd paragraph. I found the sentence ending with "(Sect 3)" a bit confusing. It is ambiguous what "(Sect 3)" refers to. Perhaps delete and then change the following sentence to read "Section 3 describes the two modelling approaches and additionally illustrates …" This makes it clearer what each section is about.

Thanks. This is updated in the revised version, Section 1 (Introduction; lines 103–106), only slightly different from your suggestion: "Section 3 describes the two modelling approaches and elaborates on the XAI method used to interpret the patterns identified by the deep learning model."

**MinC6**

> Table 1. Units should be in roman not italic font by convention.

Thanks. This is fixed.

**MinC7**

> p5, section 3.1, lines 6-7. ".. are standardized by considering the 74 levels altogether, prior training". I do not understand what this means. Please explain.

We replaced the previous, confusing description by the following more detailed description of the standardization process: "Prior to training, the input variables are standardized. For each of the atmospheric variables v , the mean mu_v and standard deviation sigma_v are calculated over all 74 model levels together, but separately for each of the nine variables."

See Section 3.1 (Deep Learning Approach; lines 154–157).

**MinC8**

> P6, line 1. I guess you are using the implementation of DeepSHAP from Lunberg available on GitHub? If so, perhaps say so explicitly / provide a reference. The footnote on p8 mentions DeepExplainer.

We have already referred to Deep SHAP and cited Lundberg (2017) in Section 3.3 (line 170). However, we have revised parts of the text and added a footnote to improve readability (lines 170, 183).

**MinC9**

> Figure 2 caption "The coloured areas highlighted the 50% quantiles." This is a bit unclear. Do you mean the shaded are shows the interquartile ranges (25%-75%)? If so, is this of the mean SHAP values calculated at each cell point, or is it the interquartile range of all SHAP values across all cells?

Sorry for the confusion. Like you suspect, the shaded areas show the interquartile range (25% - 75%) of the data. We corrected this in the revised version in the captions of Figs. 1, 2, 3, 4. Regarding your second question it is the second option (interquartile range of scaled SHAP values across all cells).

**Reply to anonymous Referee 2 (RC2)**

> The authors present a study which uses machine learning, lightning observations and reanalysis data to find which atmospheric variables are most likely linked to the occurrence of lightning. The idea presented in the paper is good, however, the overall presentation, structure and language are making it difficult to follow the authors' thoughts and scientific results. I would also encourage the authors to be clearer about the possible applications they envisage for this work. Is it to help in the formulation of new lightning parameterisation schemes for numerical weather prediction models? Or is this a system that is meant to be used with data from sounding and/or numerical models to make predictions on lightning?

We appreciate the concise feedback, which aligns well with the more detailed feedback by Reviewer 1.

Like you stated in the first sentence of your feedback, the study aims to identify which atmospheric patterns/variables are most likely associated with the occurrence of lightning by using explainable artificial intelligence to uncover the inner workings of a high-performance machine learning model.

Resulting applications e.g. are:

- Applying the methodology to regions where studies are scarce can accelerate scientific discovery in these areas and improve understanding of atmospheric processes.

- Existing models for lightning often require different parameterizations for ocean and land. The presented methodology might be used for gaining a more holistic understanding of the underlying atmospheric processes.

- The methodology itself is agnostic to lighting and can also be applied to other weather phenomena.

In the revised version we have comprehensively updated various parts of the manuscript. The most relevant parts in this context are updates in the abstract (lines 3–19, 22–26), Section 1 (Introduction; lines 45–46, 73–96; 98–109), and Section 5 (Discussion and Conclusions; 359–362; 380–382) to be clearer about the study's goals. Additionally, we have heavily restructured and reworked Section 3 (Methods; lines 145, 176–194) and Section 4 (Results; 196–199, 205–208; 215–235; 237–239; 240-247; 266–357) to improve the overall presentation.

**Additional changes**

- Typos in lines 20, 37–38, 47–48.

- Smaller improvements of language in lines 58, 60, 64–67, 69, 118–121, 125–129, 131–132, 138, 149, 162–163, 166–167, 173–176, 202, 211–213, 248, 250–251, 257, 363–364, 368, 371–376, 423–424.

- Replacing "Sect." by "Section" in lines 112–113, 146–147, 201, 209, 366.

- Replacing "Tab." by "Table" in lines 136, 164.

- Replaced wrong reference to Section 5 by correct reference to Section 3.1 in line 171.

- Replaced "Explainable AI" by "XAI" in line 390.

- Added our thanks to the reviewers to the acknowledgements in lines 425–426.